# Rapid Progression of Cutaneous Lymphoma Following mRNA COVID-19 Vaccination: A Case Report and Pathogenetic Insights

**DOI:** 10.3390/vaccines13070678

**Published:** 2025-06-25

**Authors:** Berenika Olszewska, Anna Zaryczańska, Michał Bieńkowski, Roman J. Nowicki, Małgorzata Sokołowska-Wojdyło

**Affiliations:** 1Department of Dermatology, Venereology and Allergology, Faculty of Medicine, Medical University of Gdańsk, 80-210 Gdańsk, Polandrnowicki@gumed.edu.pl (R.J.N.); mwojd@gumed.edu.pl (M.S.-W.); 2Department of Dermatology, Venereology and Allergology, University Clinical Centre, 80-214 Gdańsk, Poland; 3Department of Pathomorphology, Medical University of Gdańsk, 80-210 Gdańsk, Poland; michal.bienkowski@gumed.edu.pl

**Keywords:** mycosis fungoides, SARS-CoV-2, mRNA vaccine, COVID-19, cutaneous lymphomas, side effects

## Abstract

**Background:** Reports of primary cutaneous lymphomas (CLs) following COVID-19 vaccines are extremely rare. Nevertheless, clinicians should be aware of a potential association between these events. Here, we report a case of the development and rapid progression of mycosis fungoides (MF) with lymph node involvement after COVID-19 vaccination. **Case presentation:** A 75-year-old female developed disseminated plaques and patches shortly after receiving the first dose of the SARS-CoV-2 mRNA vaccine. Within one month following the second dose of the mRNA vaccine, she additionally experienced rapid progression, leading to the development of tumors and inguinal lymphadenopathy. Blood and visceral involvement were excluded. The clinicopathological findings were consistent with the diagnosis of MF, and systemic methotrexate with topical treatment was implemented, resulting in remission of the lesions. **Conclusions:** The presented case of the development and rapid progression of MF after the SARS-CoV-2 mRNA vaccine raises the question of the possible immunomodulatory or oncomodulatory effects of mRNA vaccines. It prompted us to conduct a review outlining the mechanisms potentially causing the mRNA vaccine-associated CLs. We have performed an extensive literature search to determine an explanation for the observed phenomenon. Accumulated evidence suggests a link between CL occurrence and immunization with an mRNA vaccine. The proposed hypothesis revolves around shared signaling pathways that are enhanced by SARS-CoV-2 mRNA vaccines, thus driving the pathogenesis of MF. We want to raise clinicians’ attention to the rare side effects of COVID-19 vaccines and emphasize the need for thorough monitoring of patients with altered immunity in the course of various lymphoproliferative disorders.

## 1. Introduction

The COVID-19 pandemic led to the introduction of novel mRNA-based vaccines against SARS-CoV-2 infection. The two currently approved mRNA vaccines are Moderna’s mRNA-1273 and Pfizer-BioNTech’s BNT162b2. Both vaccines have shown high efficacy against SARS-CoV-2 infection and are generally considered safe. The vast majority of reported cutaneous side effects are mild local reactions, mostly occurring at the injection site. However, rare cutaneous adverse events such as acute generalized exanthematous pustulosis (AGEP), livedo racemosa, urticaria, and primary cutaneous lymphomas have been reported [1,2,3,4,5]. Cutaneous T-cell lymphomas (CTCLs), with the predominant subtype mycosis fungoides (MF), are characterized by cutaneous infiltration of malignant, skin-homing T lymphocytes. The clinical course of MF varies from indolent, where the immune system effectively controls tumor growth, to aggressive forms with widespread cutaneous and extracutaneous involvement. The exact pathogenesis of CTCLs remains unknown, but the evasion of immune surveillance leading to MF progression appears to be dependent on the tumoral microenvironment [6,7]. In this context, the onset or exacerbation of CLs following COVID-19 vaccination raises the question of whether mRNA-based immunization may impair cancer immunosurveillance mechanisms. Several cases of cutaneous T-cell lymphoproliferative disorders have been reported after the administration of COVID-19 vaccines, suggesting a possible link between these events [2,3,4,5]. In this paper, we present a case of a 75-year-old patient who developed MF and subsequently experienced rapid disease progression following immunization with a COVID-19 vaccine (Comirnaty, Pfizer-BioNTech COVID-19 vaccine, New York, NY, USA). This case prompted us to review the existing literature and propose a possible pathogenetic explanation for the observed phenomenon.

## 2. Case Presentation

A 75-year-old female with a medical history of MF presented to the Department of Dermatology, Venereology and Allergology in September 2021 with widespread patches, infiltrative lesions, and a rapidly growing tumor on the leg. Comorbidities included systemic arterial hypertension, chronic heart failure NYHA class II, and atrioventricular block with a cardiac resynchronization therapy pacemaker (CRT-P).

The first erythematous itchy patches and single plaques appeared in 2019 and were confined to the trunk, covering less than 10% of the body surface area (BSA). No lymphadenopathy was observed. Histopathological examination of the patches performed in 2019 suggested eczema/psoriasis. The patient was treated successfully with topical therapies and acitretin at an outpatient clinic, maintaining stable disease. However, within a week after receiving the first COVID-19 vaccine dose on 2 June 2021, she developed multiple disseminated erythematous patches and infiltrated plaques. Histopathological examination of the skin lesion suggested MF. Furthermore, within one month following the second COVID-19 vaccination performed on 8 July 2021, she developed tumorous lesions, including a rapidly growing, widespread, and indurated tumor with a heavily exuding ulcer that covered the anterior surface of the lower left extremity (Figure 1). Physical examination revealed inguinal lymphadenopathy.

Due to MF progression, the patient was admitted on 23 September 2021 to the Department of Dermatology, Venereology and Allergology for further diagnosis of new skin lesions.

Investigations, including complete blood count and blood chemistry, revealed no abnormalities. Peripheral blood flow cytometry performed on 23 September 2021 demonstrated an absolute T-cell count of 9.6% (740/μL) and the following T-cell subsets: CD3+/CD4+ 7.4% (560/μL), CD3+/CD8+ 1.9% (150/μL), CD3+/CD56+ 0.5% (40/μL), and CD4/CD8, with a ratio of 3.9:1. Moreover, a low number of atypical T-lymphocytes was detected: CD4 T-cells (1.6%; 119/μL) with a loss of CD7 and CD26, along with a slight hypoexpression of CD3. Bone marrow immunophenotyping showed no evidence of clonality among the bone 6marrow lymphocytes. Cytological evaluation revealed no signs of infiltration by lymphoid-lineage cells. All hematopoietic lineages were appropriately represented.

Imaging studies (abdomino-pelvic ultrasound, chest X-ray, PET-CT, and CT of the chest, abdomen, and pelvis) showed no internal organ involvement, except for left inguinal lymphadenopathy.

During hospitalization, two biopsies were performed, one on the lower extremity and one on the buttock. Histopathological examination of the rapidly growing leg tumor revealed a solid infiltrate of atypical, medium-sized T lymphocytes in the stroma and in the subcutaneous tissue with focally marked epidermotropism. The immunophenotype of the tumor cells included CD3 (+), CD4 (+), CD5 (+), CD24 (+), CD7 (+), Ki67 (98%), CD30 (−), CD8 (−), and PD1 (−) (Figure 2). On the other hand, the histopathological examination. of an infiltrative plaque from the buttock, which occurred after the first vaccine dose, revealed the infiltration of CD30+ T-cells. Histology of the plaque lesion demonstrated a slightly thickened epidermis without abnormal keratosis and lymphocyte penetration. Moderate infiltration of medium-sized T-lymphocytes (CD3+, CD4+, CD8−, CD30+, and CD20−) in the periangium of the superficial plexus and single mitotic figures were present (Figure 2). Furthermore, the histopathological examination of an enlarged inguinal lymph node corresponded to lymph node involvement due to MF, with the phenotype of CD3+, CD4+, PD1−, CD7−, CD8−, CD30+, GATA3+, TIA1+, and T-bet−. Given the sum of clinicopathological findings, the patient was diagnosed with MF, stage 2B (T3 N2 M0 B0).

The patches and plaques were treated with topical steroid ointment (clobetasol propionate). Systemic antimicrobial treatment and polyurethane foam dressings were applied to the exudative ulcerated tumor on the leg. Oral methotrexate (MTX) was initially introduced at a 12.5 mg/week dose and subsequently increased to 25 mg/week with folic acid supplementation. In a relatively short period (one month), the combination of topical and systemic treatment led to the resolution of the lesions (Figure 1). Progressive re-epithelialization was observed within the ulcer on the lower extremity (Figure 1), leading to complete healing of the ulcer in 7 months. In March 2022, the patient experienced a confirmed symptomatic SARS-CoV-2 infection. A chest CT scan revealed bilateral ground-glass opacities with 75% lung involvement. After conservative treatment, the patient’s symptoms slightly improved. The treatment of MF with MTX was discontinued and replaced with peginterferon alfa-2a (90 mg weekly). The patient was subsequently hospitalized in April and May 2022 due to severe dyspnea and clinical deterioration. A follow-up CT scan demonstrated increased lung opacification (90% involvement), diffuse interstitial fibrosis, mediastinal emphysema, and a right pneumothorax. Although the patient’s condition stabilized with empiric antibiotics and oxygen supplementation, recurrent dyspnea necessitated periodic oxygen therapy. Unfortunately, within a few months, the patient died due to COVID-19-related complications.

## 3. Discussion

Several cases of primary cutaneous lymphomas have been reported following COVID-19 vaccination, including new onsets, recurrences, or disease exacerbations [2,3,4,5]. Most cases concerned primary cutaneous CD30+ lymphoproliferative disorders such as Lymphomatoid papulosis (LyP) and primary cutaneous anaplastic large-cell lymphoma (pcALCL), with rare reports of aggressive CTCLs like Sézary syndrome (SS) [2,3,4,5]. To date, only three cases of MF following SARS-CoV-2 vaccination have been reported [3,4,5], including two after mRNA-based COVID-19 vaccines. In two of these cases, early-stage MF progressed to tumor or erythrodermic stages [3,4], while another report described a new onset of CD8+ MF after mRNA vaccination [5]. Notably, all patients experienced disease progression after subsequent vaccine doses. Nevertheless, two of the three patients achieved complete remission with treatment [4,5], while the outcomes for the third case were not reported [3].

We have also reported a case of MF development and progression following mRNA-based COVID-19 vaccination, suggesting a potential etiologic correlation. Notably, the patient exhibited dramatic and rapid exacerbation of skin lesions but also lymph node involvement after subsequent doses of the mRNA vaccine. Remarkably, despite the staggering progression after the second vaccination, standard treatment resulted in remission of plaque lesions and impressive resolution of tumors. The similarity to previously reported cases of CLs raises concern about a possible association between MF and mRNA COVID-19 vaccination [2,3,4,5]. It should be emphasized that no research papers investigating the mechanism of action leading to post-vaccine CLs have been published so far. We hypothesize that repeated vaccinations may have triggered CD30 overexpression in the subset of lesions and further reduction in already exhausted T-cells and T-cell receptor (TCR) diversity, leading to impaired immune surveillance and uncontrolled tumor growth in our patient. It was already demonstrated that antigenic stimulation by mitogens and viruses can drive CD30 expression in lymphocytes [8] and can potentially contribute to T-cell exhaustion. The exhaustion of CD8+ T-cells has also been associated with repeated COVID-19 vaccination [9]. Brumfiel et al. and Panou et al. reported cases of LyP relapse after vaccinations, suggesting that the activation of CD30 may be provoked by viral stimuli [2,3]. Moreover, we agree with the speculations of Panou et al. and believe that CD30 expression and T-cell exhaustion are co-consequences of antigenic stimulation, rather than CD30 being the cause of exhaustion. 

We further suspect that the rapid progression observed in this case may result from overlapping immunologic and oncogenic signaling pathways. The vaccine-induced immune responses may amplify dysregulated pathways underlying CTCL pathogenesis, thereby contributing to disease progression. One possible explanation is that mRNA COVID-19 vaccines, while highly effective in generating antiviral immunity, might concurrently activate neoplastic T-cell clones via the stimulation of the nuclear factor kappa-light-chain-enhancer of activated B cells (NF-κB) and the signal transducer and activator of transcription 3 (STAT3) signaling pathways.

mRNA vaccines use mRNA combined with lipid nanoparticles (LNPs) to induce the expression of the SARS-CoV-2 spike (S) protein, which facilitates viral entry into the host cell. The expression of the S protein following vaccination elicits an adaptive immune response and the production of virus-neutralizing antibodies [10,11]. LNPs, particularly their ionizable lipid components, possess potent adjuvant activity, including the activation of various inflammatory pathways and the production of inflammatory cytokines [12]. As a result, the intramuscular administration of LNP-mRNA COVID-19 vaccines induces an inflammatory milieu, often leading to commonly reported injection site reactions.

Both mRNA and ionizable lipids can drive proinflammatory pathways by the activation of innate immune sensors, including various Toll-like receptors (TLRs), which are crucial in the antiviral immune response [13,14]. Most importantly, viral proteins, not specifically related to SARS-CoV 2, stimulate TLR signaling, triggering NF-κB activation and the subsequent production of numerous proinflammatory cytokines, including IL-6 [15]. Interestingly, increased epidermal expression of TLRs has been observed in MF lesions (TLR-2, TLR-4, and TLR-9) [16] and in CD30+ cutaneous lymphomas (TLR-2, TLR-4, TLR-7, and TLR-9) [17].

Furthermore, Hirsiger et al. reported a case of agranulocytosis and T-cell large granular lymphocytic leukemia following mRNA vaccination, associated with STAT3 activation via TLR stimulation and the subsequent secretion of IL-6 [18]. This crucial finding suggests that mRNA COVID-19 vaccines may stimulate the STAT3 pathway, potentially exacerbating STAT3-dependent diseases such as MF. The deregulation of signaling pathways, particularly JAK/STAT and NF-κB, is known to play an essential role in CTCL pathogenesis. Constitutive activation of STATA3 in MF tumor cells is associated with disease progression and large-cell transformation [19,20]. Both STAT3 and NF-κB play a key role in tumor cell proliferation, migration, increased survival, and resistance to apoptosis in malignant T-cells [21]. Recent evidence links the adverse effects of mRNA COVID-19 vaccines not only to the LNP–mRNA platform but also to the biological effects of the S protein [21]. The “spike hypothesis” suggests that the synthesized S protein or its peptide fragments may enter the circulation following vaccination, triggering inflammatory signaling and cytokine production, potentially resulting in adverse events (AEs) [11].

Numerous studies have demonstrated that the SARS-CoV-2 S1 spike protein mediates systemic inflammation by stimulating the production of inflammatory cytokines (e.g., TNF-alfa, IL-6, and IFN gamma) and chemokines, activating both the ERK1/2 MAPK and NF-κB pathway via TLRs [22,23,24]. Importantly, data have shown that the spike protein alone is sufficient to trigger proinflammatory activity, independent of other viral components [25]. This supports the idea that AEs following immunization with COVID-19 mRNA vaccines are a result of the synthesized SARS-CoV-2 spike proteins that elicit cell signaling in human cells.

Taken together, we hypothesize that COVID-19 vaccines may unmask subclinical lymphoproliferation and promote disease progression by inducing a tumor-promoting microenvironment. This oncomodulatory effect likely involves signaling crosstalk between cytokines, NF-κB, and the STAT3 pathway. COVID-19 mRNA vaccines activate NF-κB, which stimulates the production of various cytokines such as IL-6. The binding of IL-6 to its receptors subsequently activates STAT3 oncogenic signaling [26], potentially promoting the cutaneous lymphoma development or progression in susceptible individuals [27,28]. Therefore, COVID-19 mRNA vaccines seem to have great potential to bias the STAT3 and NF-kB-dependent diseases. A schematic summary of the proposed mechanism is presented in Figure 3.

However, it should be noted that the incidence of MF exacerbations or new onsets following vaccination is rare. We suspect that individuals with CTCL may be susceptible to COVID-19 vaccine-induced adverse events due to genetic or environmental factors. We speculate that certain genomic alterations, including overactive T-cell and cytokine receptor signaling, JAK/STAT, NF-κB signaling, or other related mutations, promote constitutive activation of transcription pathways, and that mRNA COVID-19 vaccines may trigger clonal proliferation in such cases.

## 4. Conclusions

In conclusion, although mRNA vaccines are highly effective against SARS-CoV-2 infection, they may cause side effects in susceptible populations. Although reports linking COVID-19 vaccines to the development or exacerbation of CLs are rare, they require thorough investigation. The presented case supports an etiological correlation between the Pfizer-BioNTech SARS-CoV-2 mRNA vaccine and MF progression. We propose that MF following COVID-19 immunization results from crosstalk between cytokines and signaling pathways induced by vaccine components, leading to oncomodulation rather than direct oncogenesis.

Given that undetermined predisposing factors likely contribute to these cutaneous adverse reactions, further detailed studies are needed to ensure vaccine safety, especially in patients with altered immunity or lymphoproliferative diseases.

## Figures and Tables

**Figure 1 vaccines-13-00678-f001:**
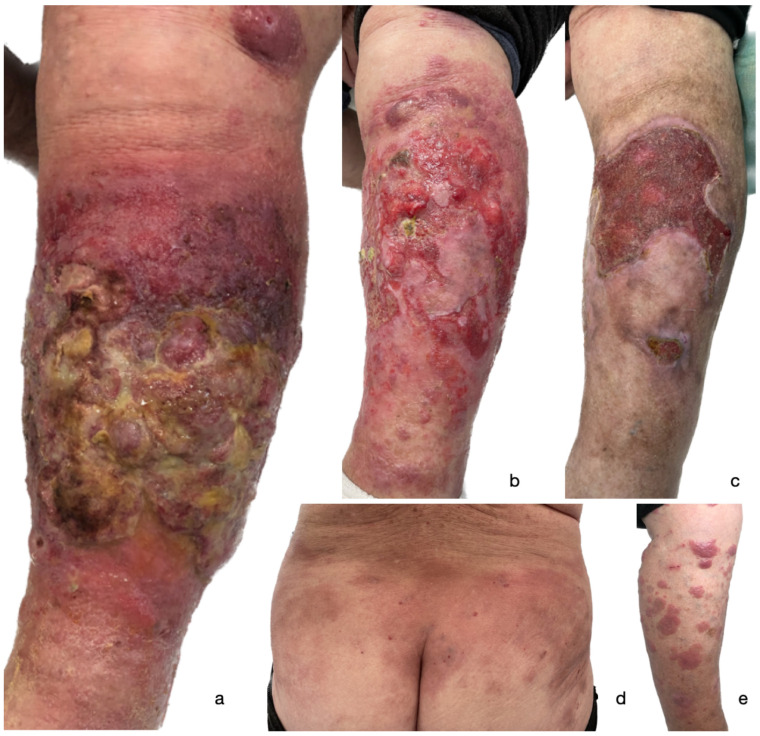
A clinical presentation of a widespread ulcerated tumor on the left lower extremity of the patient following 2nd dose of BNT162b2 COVID-19 mRNA vaccine (**a**); gradual improvement of the lesion following one month of treatment (**b**); partially healed ulcer five months following initial presentation (**c**). Patches and plaques on the trunk (**d**) and right lower extremity (**e**) that occurred after 1st dose of BNT162b2 COVID-19 mRNA vaccine.

**Figure 2 vaccines-13-00678-f002:**
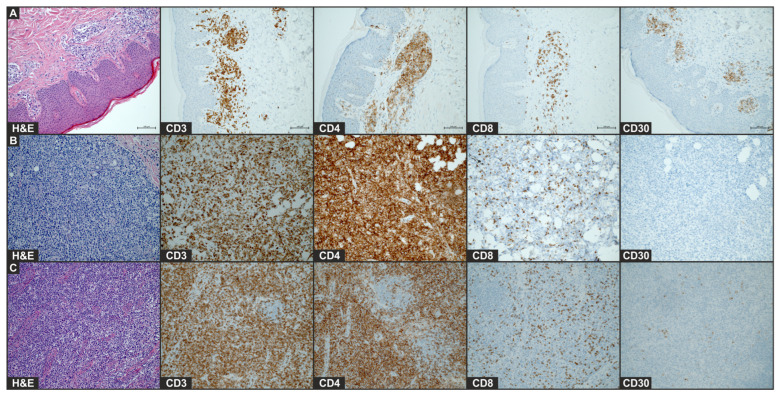
Histology of plaque lesion (**A**), tumor (**B**), and enlarged lymph node (**C**) after COVID-19 mRNA vaccine. Images of H&E staining and immunohistochemical staining for CD4, CD8, and CD30 (×20) of the biopsy of initial plaque that occurred after first vaccination and leg tumor and lymph node that developed after second vaccination.

**Figure 3 vaccines-13-00678-f003:**
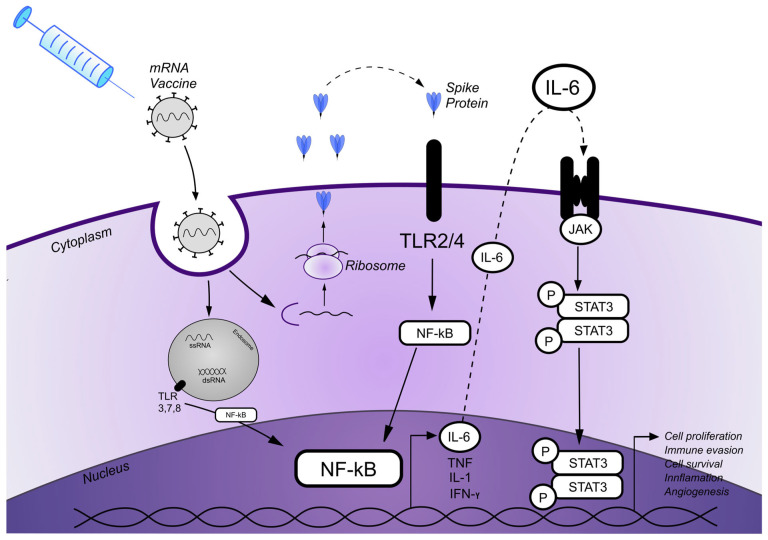
A schematic diagram illustrating the proposed patomechanism of MF onset/progression induced by the SARS-CoV-2 mRNA vaccine. mRNA carried in the mRNA vaccine enters cells by endocytosis. After release from the endosome, mRNA is translated into proteins by ribosomes that are also secreted extracellularly. Spike (S) proteins bind to Toll-like receptors (TLRs), which are expressed on the cell surface, and subsequently activate the nuclear factor kappa-light-chain-enhancer of activated B cells (NF-κB) pathway and its target genes, resulting in the production of proinflammatory cytokines. Single-stranded RNA (ssRNA) and double-stranded RNA (dsRNA) vaccines, likewise, bind to TLRs in the endosome and activate the NF-κB signaling pathway and target genes. Various cytokine secretions, including IL-6, are initiated via the NF-κB pathway. IL-6 subsequently binds to its surface receptors, which results in activation of the JAK/STAT3 signaling pathway, leading to the transcription of STAT3 target genes that potentially promote cell proliferation, inflammation, cell survival, and angiogenesis.

## Data Availability

Data is contained within the article.

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
