# Peer review of "Rapid Progression of Cutaneous Lymphoma Following mRNA COVID-19 Vaccination: A Case Report and Pathogenetic Insights"

_vaccines, 2025, doi:10.3390/vaccines13070678_

Round 1

Reviewer 1 Report

Comments and Suggestions for Authors

Please provide complete medical history and current medications at the first patient visit.

Please provide the date of the second covid vaccine.

Explain why the second dose was administrated despite what had been observed after the first dose.

Please provide the absolute counts of T cells and different subsets of T cells in the peripheral blood. Also provide the exact date of this phenotyping (how many days after the second dose of vaccine).

The sentence "Peripheral blood flow cytometry presented low CD4 T cells (1.6%) with loss of CD7 and CD26 , along with slight hypoexpression of CD3" is not very clear. I suppose it refers to a clonal population of CD4 T cells present in the blood but the way it is written could suggest that the patient suffers from severe CD4 T cell lymphophenia. Please clarify.

I am bit confused about the chronology of the biopsy of the infiltrative plaque from the buttock. Was it performed after the first of the second dose of vaccine?

In the discussion (line 147), the sentence "further reduction of already exhausted T cells diversity" is not clear. It mixes two different concepts (diversity on the one hand and exhaustion on the other). Please clarify. To my knowledge, neither exhaustion nor diversity have been studied in this patient.

Line 149, another reference should be given. Reference 8 is a review paper which does not directly address this issue. Why not by example Ellis TM et al. 1993

Line 149 again. "It was already demonstrated that antigenic stimulation by mitogens and viruses can drive the CD30 expression on lymphocytes [8], 149 potentially contributing to T- cell exhaustion". What does it mean exactly? Do the authors postulate that CD30 expression is responsible for exhaustion? Or do they postulate that CD30 expression and exhaustion are two independent consequences of antigenic stimulation. Please clarify.

Line 150 : the authors state that reference 9 describes "Depletion of both CD4+ and CD8+ T-cells 150 with repeated COVID-19 vaccination". This is totally false. This article did not show any depletion of CD4 or CD8 T cells but in some patients a slightly increased proportion of PD-1 positive CD8 T cells. The authors of reference 9 clearly state : "No differences in CD4 + T cells values between the different groups were observed". It is particulary important to correct this false assertion since the patient described in the manuscript might present with CD4 lymphopenia (see above)

Line 152, the two papers cited evoke CD30 stimulation but not overexpression.

Line 172, the term "antigen" refers to interaction with immunoreceptors (i.e. TCR or BCR). It would be more appropriate to write "viral proteins stimulate TLR..."

Line 174, it is maybe important to remind that reference 15 is not specifically related to Sars CoV2 proteins.

Line 190. Trougakos et al. (reference 11) do not postulate that the vaccine induces a cytokine storm. They evoke the cytokine storm as a consequence of Sars CoV2 infection, not of the vaccine. Read the paper.

The mechanism suggested by the authors involves a central role for the Spike protein. It would therefore be important to discuss the role of Sars CoV2 infection itself (where exposure to the Spike protein is considerably greater) in the development or exacerbation of lymphoproliferative phenomena. It would also be interesting to discuss the evolution of the incidence of these lymphomas since 2021. If the authors' hypothesis is correct, this incidence should be increased, which is not the case according to my analysis.

Author Response

Thank you very much for taking the time to review our work and for your valuable suggestins. We have revised the manuscript according to the comments, please find revisions marked in red in uploaded "revised manuscript". 

Comments 1: Please provide complete medical history and current medications at the first patient visit.

Response 1: We have provided the information- line 65-67

Comments 2: Please provide the date of the second covid vaccine.

Response 2: We have provided the information- line 76 

Comments 3: Explain why the second dose was administrated despite what had been observed after the first

Response 3:At the time of the skin lesions onset, the patient was treated and diagnosed at a different healthcare facility (outpatient clinic) located in her place of residence, where she also received her vaccinations. Therefore, it is difficult for us to determine the rationale behind the administration of the second vaccine dose. However, according to CL guidelines, including Cutaneous Lymphoma Foundation and EORTC-CLTF, patients with cutaneous T-cell lymphomas ( Mycosis Fungoides) are prioritized for complete COVID-19 vaccination. No guidelines contraindicate continuation of the vaccination schedule following disease exacerbation. It is emphasized that the benefits of vaccination outweigh the potential risk of adverse effects, including exacerbation of MF. 

Comments 4: Please provide the absolute counts of T cells and different subsets of T cells in the peripheral blood. Also provide the exact date of this phenotyping (how many days after the second dose of vaccine).

Response 4: We have provided the information- line 91-97 

Comments 5: The sentence "Peripheral blood flow cytometry presented low CD4 T cells (1.6%) with loss of CD7 and CD26 , along with slight hypoexpression of CD3" is not very clear. I suppose it refers to a clonal population of CD4 T cells present in the blood but the way it is written could suggest that the patient suffers from severe CD4 T cell lymphophenia. Please clarify.

Response 5: Thank you for this comment, we agree with this comment. We have clarified the sentenc- line 95-97

Comments 6: I am bit confused about the chronology of the biopsy of the infiltrative plaque from the buttock. Was it performed after the first of the second dose of vaccine?

Response 6: We have clarified the sentence. The biopsy was performed after second dose of vaccine.  However, lesion on buttock occured after first vaccination, in contrast to the leg tumor which occured after second vaccination- line 104-105

Comments 7: In the discussion (line 147), the sentence "further reduction of already exhausted T cells diversity" is not clear. It mixes two different concepts (diversity on the one hand and exhaustion on the other). Please clarify. To my knowledge, neither exhaustion nor diversity have been studied in this patient.

Response 7: Thank you for this comment. We agree, this was our linguistic mistake. The sentence lack TCR . We have corrected it - line 167 

You are right — no PCR testing was performed. Our hypothesis is based  directly on the pathomechanism of primary cutaneous lymphomas, including MF. MF arises from malignant transformation of a single or a few T-cell clones, which initially expand within the skin and reduce the healthy T-cell repertoire. The malignant clones dominate, and the diversity of TCR sequences is reduced. We speculate that exposure to antigenic stimuli  for example viral elements  (s protein) might lead to activation and exhaustion of the already limited non-malignant T cells which in consequence might contribute to either reccurance, develepment or progression.

 Comments 8:Line 149, another reference should be given. Reference 8 is a review paper which does not directly address this issue. Why not by example Ellis TM et al. 1993

Response 8: We have made the correction in accordance with the suggestion- reference was changed - line 170

Comments 9: Line 149 again. "It was already demonstrated that antigenic stimulation by mitogens and viruses can drive the CD30 expression on lymphocytes [8], 149 potentially contributing to T- cell exhaustion". What does it mean exactly? Do the authors postulate that CD30 expression is responsible for exhaustion? Or do they postulate that CD30 expression and exhaustion are two independent consequences of antigenic stimulation. Please clarify.

Response 9: Our speculations are in line with Panu hypothesis. We believe that CD30 expression and T-cell exhaustion are co-consequences of antigenic stimulation, rather than CD30 being the cause of exhaustion. - line 174-176

Comments 10: Line 150 : the authors state that reference 9 describes "Depletion of both CD4+ and CD8+ T-cells 150 with repeated COVID-19 vaccination". This is totally false. This article did not show any depletion of CD4 or CD8 T cells but in some patients a slightly increased proportion of PD-1 positive CD8 T cells. The authors of reference 9 clearly state : "No differences in CD4 + T cells values between the different groups were observed". It is particulary important to correct this false assertion since the patient described in the manuscript might present with CD4 lymphopenia (see above)

Response 10: Thank you for pointing this out. We agree and have revised the statement- line 171-172

Comments 11: Line 152, the two papers cited evoke CD30 stimulation but not overexpression.

Response 11: We have modified this section in accordance with the suggestion- line 173 

Comments 12: Line 172, the term "antigen" refers to interaction with immunoreceptors (i.e. TCR or BCR). It would be more appropriate to write "viral proteins stimulate TLR..."

Response 12: We have revised this term in accordance with the suggestion- line 195

Comments 13: Line 174, it is maybe important to remind that reference 15 is not specifically related to Sars CoV2 proteins.

Response 13: We have revised and clarified this sentence- line 197

Comments 14: Line 190. Trougakos et al. (reference 11) do not postulate that the vaccine induces a cytokine storm. They evoke the cytokine storm as a consequence of Sars CoV2 infection, not of the vaccine. Read the paper.

Response 14:  Thank you for your remark — we have revised the text to make this clearer- line 214

Comments 15: The mechanism suggested by the authors involves a central role for the Spike protein. It would therefore be important to discuss the role of Sars CoV2 infection itself (where exposure to the Spike protein is considerably greater) in the development or exacerbation of lymphoproliferative phenomena. It would also be interesting to discuss the evolution of the incidence of these lymphomas since 2021. If the authors' hypothesis is correct, this incidence should be increased, which is not the case according to my analysis.

Response 15:

Thank you for this thoughtful suggestion. While we agree that the role of SARS-CoV-2 infection in lymphoproliferative processes is a valuable area of exploration, we believe that such a discussion lies beyond the scope of a single case report. Our primary aim was to highlight a rare clinical event and propose hypothesis-generating mechanisms based on immune activation pathways possibly triggered by spike protein.

There is indeed a substantial body of literature addressing lymphoproliferative complications following SARS-CoV-2 infection. However, expanding this discussion here would likely detract from the case-specific focus and introduce redundant material. We have already covered the topic of incidence of cutaneous lymphomas in our review (Olszewska et al., Front Med, 2024). In that review, we  found no definitive trend indicating increased incidence of cutaneous lymphomas, supporting the Reviewer’s observation.

In this manuscript, we emphasize that the proposed mechanism remains speculative and is intended to stimulate further inquiry. We consider the most plausible explanation for the observed flare to be the alteration of the skin immune microenvironment and activation of intracellular signaling pathways in susceptible individuals. As we note, such adverse events are exceedingly rare, and likely modulated by individual genomic or environmental susceptibility factors.

We hope you will agree that, for clarity and precision, our case report is best when  focused on the direct clinical observation and hypothesized pathway, without broadening into  well-reviewed topic of SARS-CoV-2 infection in lymphoproliferative diseases. 

Reviewer 2 Report

Comments and Suggestions for Authors

Vaccines are an important tool in the fight against viral infections. Unfortunately, we should not overlook the possible side effects in susceptible populations. Describing clinical cases associated with vaccination helps identify possible connections between vaccination and complications. Here is a report of the development and rapid progression of mycosis fungoides with lymph node involvement after vaccination against COVID-19.

Some comments are listed below:
Line 59.Please add information about the date of the patient's visit.
Lines 59–62. Please add a detailed description of the 2019 erythematous lesions. Is there a connection between these skin lesions and cutaneous lymphoma after vaccination? Could this be a relapse after vaccination?
Line 64. "Histopathological examination of the patch suggested eczema/psoriasis." Clarify that this information is about the 2019 disease.
Line 67. Please provide the date of vaccination and the date the symptoms began.
Line 81. Describe the bone marrow immunophenotyping test in more detail.

Author Response

Thank you very much for taking the time to review our work and for your valuable comments. We have revised the manuscript according to the commnerts, please find revisions marked in red in uploaded manuscript. 

Comments 1: Line 59.Please add information about the date of the patient's visit.

Response 1: the date of the patient's visit was added - line 64, additional information added in line 80-82 for clarity 

Comments 2: Lines 59–62. Please add a detailed description of the 2019 erythematous lesions. Is there a connection between these skin lesions and cutaneous lymphoma after vaccination? Could this be a relapse after vaccination?

Response 2:. detailed description of lesions was added- line 68

The patient was initially treated at other facility in a dermatology outpatient clinic and was admitted to the Department of Dermatology in September 2021 with fully developed skin lesions consistent with mycosis fungoides (MF). Unfortunately, we did not see the patient at the onset of her disease therefore, our assessment in this regard is limited. However, at the time of disease onset the patient was diagnosed based on the clinical presentation and histopathological examination with psoriasis. MF is a rare disease, and in its early stages, the clinical presentation can mimic other benign dermatoses, such as allergic eczema or psoriasis. In the early phases, the histopathological features are often non-specific as well, making the diagnosis challenging even for experienced pathologists. In this particular case, it is difficult to provide a definitive answer.

Comments 3: Line 64. "Histopathological examination of the patch suggested eczema/psoriasis." Clarify that this information is about the 2019 disease.

Response 3: we have clarified the information  -line 70 

Comments 4: Line 67. Please provide the date of vaccination and the date the symptoms began.

Response 4: The date of vaccination was added (line 73), however we can not provide the exact date of  symptoms begining. According to the patient’s medical history, nodular lesions appeared within a month after the second vaccine dose, the patient was unable to provide the exact date. 

Comments 5:Line 81. Describe the bone marrow immunophenotyping test in more detail.

Response 5: bone marrow immunophenotyping detailed result was added-  line 97-100

Reviewer 3 Report

Comments and Suggestions for Authors

This is a case report of progression of cutaneous lymphoma after mRNA COVID-19 vaccination and a review of related literature. As such, it is of interest and warrants publication. However, the title word “double-edged sword” is too dramatic and should be deleted wherever mentioned. The report is just as good without. Even with the literature review the progression of cutaneous lymphomas must be very rare (not that the incidence is “relatively low”) because of the paucity of reports.  

The Discussion contains a lot of speculation. In this case I would not urge to shorten it.

Author Response

Comments 1: However, the title word “double-edged sword” is too dramatic and should be deleted wherever mentioned.

Response 1: Thank you very much for taking the time to review our work and for your valuable feedback. Our original title was intended to draw attention to the fact that COVID-19 vaccines are generally beneficial in preventing COVID-19 infection, yet may, in certain circumstances—such as the lymphoproliferative disorder presented—have unfavorable effects. We have now revised the title in line with your suggestion. 

Changed Title : Rapid Progression of Cutaneous Lymphoma Following mRNA COVID-19 Vaccination: A Case Report and Pathogenetic Insights.

We have modified the manuscript text in accordance with your recommendations - double-edged sword deleted. Line: 30, 255, 

Comments 2:  Even with the literature review the progression of cutaneous lymphomas must be very rare (not that the incidence is “relatively low”) because of the paucity of reports.  

Response 2: We have modified the manuscript text in accordance with your recommendations- line 234

Round 2

Reviewer 1 Report

Comments and Suggestions for Authors

Most of my concerns have been addressed. The manuscript can be published. My only concern relates to the peripheral blood phenotyping

  1. The authors mentioned a abnormal subset (CD7 low and CD26 low). This populations is usually considered to be clonal. Nevertheless, the authors state that "cytometry demonstrated no evidence of clonality". What was the technique used to rule out clonality. What is the explanation of the "absence of clonality" since 1.6% abnormal T cells (presumably clonal) are present.
  2.  The way to express the absolute values of lymphocytes is absolutely not typical. Please use number of cells per microliter. 

Author Response

Comments 1: The authors mentioned a abnormal subset (CD7 low and CD26 low). This populations is usually considered to be clonal. Nevertheless, the authors state that "cytometry demonstrated no evidence of clonality". What was the technique used to rule out clonality. What is the explanation of the "absence of clonality" since 1.6% abnormal T cells (presumably clonal) are present.

Response 1: Thank you for this comment. We agree that T cells with this aberrant immunophenotype might be associated with Sézary syndrome and mycosis fungoides. However, an aberrant phenotype alone does not constitute definitive evidence of clonality. Importantly, the loss of CD7 or CD26 may also occur in reactive T-cell populations, for example eczema or psoriasis and in low levels in healthy individuals.

In our case, the number of aberrant CD4⁺ T cells (CD26⁻CD7⁻) was only 119/ μL (1.6%), which is insignificant, defining the stage of blood involvement as B0.  In cases of B0 blood involvement with a low numer of abberant t-cells, molecular clonality testing (TCR gene reagrangement) is not required according to current EORTC/ISCL guidelines. 

As clarified by the laboratory, the provided description— “cytometry demonstrated no evidence of clonality” — referred to the absence of a dominant immunophenotypic clone based on flow cytometric analysis of CD4, CD7, CD26 expression. According to MF guidelines, T-cell clonality should ideally be verified via TCR gene rearrangement analysis to confirm clonality in cases when the number of aberrant CD4⁺CD26⁻ or CD4⁺CD7⁻ T-cells approaches or exceeds thresholds for B1/B2 staging. At the time of the examination (2021), the laboratory did not perform molecular analysis of T-cell receptor gene rearrangements (PCR or NGs). Therefore, the absence of clonality was interpreted within the limits of immunophenotypic resolution. Thank you for your valuable comment, for clarity we revised this statement.  The revisions in the text are highlighted in yellow (lines 91–95).

Comments 2: The way to express the absolute values of lymphocytes is absolutely not typical. Please use number of cells per microliter.

Response 2: Thank you for the suggestion regarding units of measurement, we agree with the comment. The values were initially expressed in gigaliters (G/L), a standard unit in hematology reports. We have corrected the values. The revisions in the text are highlighted in yellow (lines 91–95).